# Enhanced heterogeneous ice nucleation by special surface geometry

Yuanfei Bi[1], Boxiao Cao[1] & Tianshu Li[1]

The freezing of water typically proceeds through impurity-mediated heterogeneous nucleation. Although non-planar geometry generically exists on the surfaces of ice nucleation centres, its role in nucleation remains poorly understood. Here we show that an atomically sharp, concave wedge can further promote ice nucleation with special wedge geometries. Our molecular analysis shows that significant enhancements of ice nucleation can emerge both when the geometry of a wedge matches the ice lattice and when such lattice match does not exist. In particular, a 45° wedge is found to greatly enhance ice nucleation by facilitating the formation of special topological defects that consequently catalyse the growth of regular ice. Our study not only highlights the active role of defects in nucleation but also suggests that the traditional concept of lattice match between a nucleation centre and crystalline lattice should be extended to include a broader match with metastable, non-crystalline structural motifs.

[1] Department of Civil and Environmental Engineering, George Washington University, 800 22nd Street NW, Washington, DC 20052, USA. Correspondence and requests for materials should be addressed to T.L. (email: tsli@gwu.edu).

Surface roughness has been known to promote nucleation. In fact, it is a common experimental practice to scratch surface, to induce nucleation on surface irregularities such as grooves and pits. The active role of these surface irregularities on nucleation, in particular for a concave cavity, is well described by the heterogeneous classical nucleation theory (CNT) through a simple geometric argument[1]: a concave cavity reduces the volume of critical nucleus further than a flat or a convex surface, thus favouring the nucleation of a new phase. By the same argument, the theory predicts that the nucleation barrier monotonically decreases as cavity becomes sharper, that is, with a smaller tip radius.

Although the theory works well for the nucleation of gas and liquid, crystallization of solids on rough surface is often found to exhibit much more complex behaviours. For example, ice nucleation is found to be significantly promoted by surface irregularities on hematite particles[2] and $BaF_2$ (111) surface[3]. Very recent studies also provide direct experimental evidences that the acute wedges of mica[4] and surface defects of potassium-rich feldspars[5,6] are effective nucleation sites for ice crystallization from vapour. In contrast, droplet freezing experiments show surface roughness has a negligible effect on ice nucleation on superhydrophobic surfaces[7]. Such insensitivity is also reported on silicon, glass and mica substrates[8]. Importantly, experiments also suggest that the role of surface roughness could be coupled with other factors such as surface chemistry[9].

The complexity in the surface-irregularity-induced crystallization is primarily due to the crystalline nature of solids. If a rough surface disrupts the crystalline ordering of solids, the nucleation of solids may become unfavourable. For example, simulations show that atomically rough graphitic surface suppresses density layering of water and inhibits heterogeneous ice nucleation[10]. Similarly, if there exists a certain structural match between surface irregularity and the lattice of solids, crystallization may be significantly promoted. Indeed, molecular crystals are selectively crystallized along the line of wedges obtained through crystal cleavage[11–13]. Simulations further show that there exists an optimal wedge angle where the nucleation rate of Lennard–Jones crystal becomes orders of magnitude higher in groove than on flat surface[14].

Here we investigate heterogeneous ice nucleation within an atomically sharp, concave wedge through forward flux sampling (FFS) method[15] and the mW water model[16]. We find the enhancement of ice nucleation within a concave wedge relative to planar surface only occurs under special wedge geometries. When wedge structurally matches the orientations of specific ice lattice planes, it can greatly enhance not only ice nucleation rate but also the propensity of a specific ice structure, thus allowing potentially controlling polymorph selection of ice. Surprisingly, when wedge does not match ice lattice, we find it may also significantly promote ice nucleation, as in the 45° wedge. The unexpected rate enhancement is found attributed to the favourable formation of metastable topological defects upon geometrical constraints exerted by wedge. The non-ice-like structural units subsequently facilitate the growth of regular ice structure, thus accelerating ice crystallization.

## Results

### Ice nucleation rate.
The atomically sharp wedge is created by joining two graphene planes at a contact line with a wedge angle $\beta$. The relative crystalline orientation of graphene sheets is insignificant, as our previous study[17] show that the crystallinity of graphene plays no active role in heterogeneous ice nucleation with the original water–carbon interaction strength[10]. For simplicity, the two graphene planes are kept at the same crystalline orientation (see Methods for more details). A large wedge angle eventually turns a wedge into a flat surface, that is, $\beta = 180°$, whereas a very sharp wedge leads to confinement that also shifts the phase diagram of ice[18]. Therefore, we confine our ice nucleation study within a range of $\beta \in (30°, 150°)$. Figure 1a shows the calculated ice nucleation rate as a function of wedge angle $\beta$ on the basis of the mW water model[16]. Nearly for all the angle $\beta$ investigated, the wedge produces a nucleation rate higher than that on a flat carbon surface. At the first glance, this appears consistent with CNT qualitatively. A closer examination, however, shows the fundamental difference between our simulation results and the CNT prediction. First, although nucleation is generally enhanced by a wedge, the calculated nucleation rates exhibit a non-monotonic dependence on wedge angle. In contrast, CNT predicts a simple reduction of nucleation barrier or an enhancement of nucleation rate with respect to a decreasing wedge angle $\beta$. Second, there exist wedge angles, for example, 30°, 60° and 135°, where the calculated ice nucleation rate becomes virtually indistinguishable from that on flat surface. Examination of these nucleation trajectories shows that at these angles, ice indeed nucleates in the planar region of the carbon wedge, far away from the contact line of the wedge. Therefore, ice nucleation proceeds in the same manner as it does on flat graphene, regardless of wedge geometry. The absence of rate enhancement at these angles suggests that an atomically sharp wedge does not always promote crystallization.

As CNT does not provide a rational explanation of the simulation results, we examine the nature of rate enhancement at the other wedge angles. Figure 1a also shows that ice nucleation is in fact promoted significantly only when $\beta$ is around three angles, namely, 45°, 70° and 110°. At 70° and 110°, we find that the rate enhancement originates from the structural compatibility between the wedge and cubic ice $I_c$. As graphene is known to promote the formation of the basal plane of hexagonal ice $I_h$[10,17,19] (which is equivalent to the {111} plane of cubic ice $I_c$), and as the dihedral angle between two intersecting {111} planes is 70.52° (or 109.48°), the growth of unstrained cubic ice $I_c$ thus fits the wedge geometry when $\beta$ is near these angles. When $\beta$ deviates from these angles, the nucleation of strained ice crystal must also overcome an additional strain cost, which leads to a decreasing nucleation rate. A similar behaviour has also been observed in the nucleation of Lennard–Jones particles[14].

### Enhanced ice nucleation through matching ice lattice.
The enhancement at 70° and 110° can thus be understood in terms of the templating effect in a broader sense: when a nucleation agent is able to create an ordering compatible with the structure of a crystalline phase, it will then promote the nucleation of the corresponding phase; if the induced ordering could further match the crystalline structure at a higher degree, the nucleation efficiency of an agent can be even more enhanced[17]. Indeed, in carbon–water system, a single graphene plane induces layering within the interfacial water that matches the density profile of ice normal to the basal plane[10,17,19,20]. This one-dimensional density match alone leads to an enhancement of ice nucleation rate by 25 orders of magnitude at 240 K, from $1.67 \times 10^{-7} \, \mathrm{m}^{-3} \, \mathrm{s}^{-1}$ for homogeneous ice nucleation[21] to $9.34 \times 10^{18} \, \mathrm{m}^{-3} \, \mathrm{s}^{-1}$ on graphene surface[22]. When adding the second graphene so that the structural match occurs in two dimensions simultaneously, as in the 70° wedge, ice nucleation rate is further promoted by another eight orders of magnitude, yielding $8.6 \times 10^{26} \, \mathrm{m}^{-3} \, \mathrm{s}^{-1}$ at the same temperature. Similarly, adding the third dimensional match is then expected to continue boosting ice nucleation. To confirm this conjecture, we create a tetrahedral pyramid by adding the third graphene plane to the 70° wedge so that all three

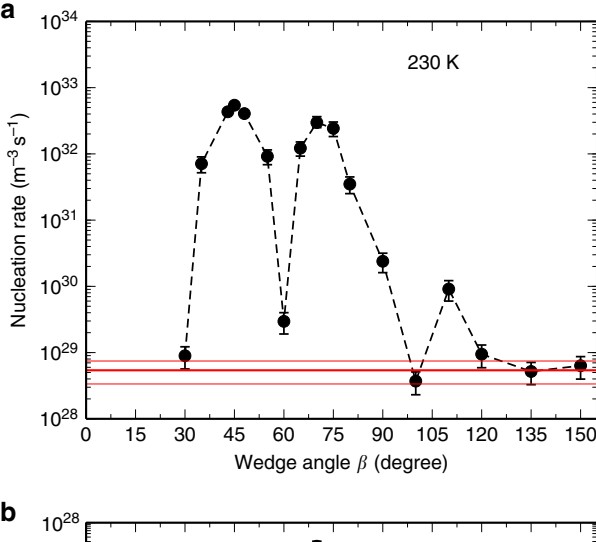

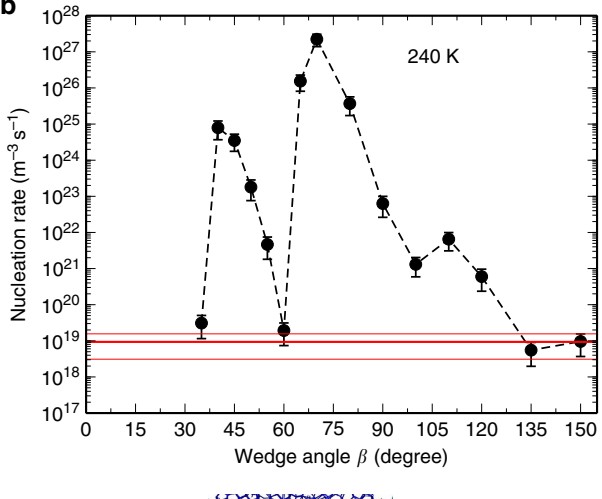

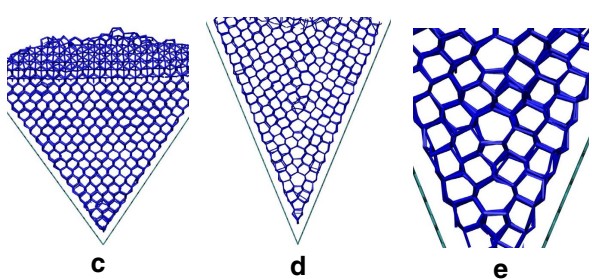

**Figure 1 | Crystallization of mW water within atomically sharp wedge.**
Calculated rate constant of heterogeneous ice nucleation within an
atomically sharp wedge as a function of wedge angle $\beta$ at (**a**) 230 K and
(**b**) 240 K. The red lines indicate the heterogeneous ice nucleation rates
(with statistical uncertainty) computed on the flat graphene plane at the
corresponding temperatures[22]. All the nucleation rates are calculated based
on FFS method, except for those obtained at 230 K for the 43°, 45° and 47°
wedges, where spontaneous nucleation occurs frequently in direct MD
simulations. In these cases, the nucleation rates are computed directly
based on multiple direct MD shootings (see Methods for details). The
statistical uncertainty of those nucleation rates calculated based on FFS is
obtained by estimating both the variance in the binomial distribution of the
number of configurations collected at each interface and the landscape
variance in the starting configurations at each previous interface[42].
(**c**,**d**) Side view of the fully crystallized ice at 230 K within the 70° and the
45° wedges, respectively. (**e**) Zoom in of **d**, to highlight the topologically
defective structure. Water molecules and carbon atoms are represented by
blue and green, respectively.

graphene planes now are able to match three intersecting {111}
planes simultaneously. When filled with water inside, this
tetrahedral pyramid wedge is found to lead to spontaneous ice
crystallization within 2 ns in direct molecular dynamics (MD)
simulation at 240 K. In fact, the nucleation efficiency for
tetrahedral pyramid is so high that one has to raise the
temperature significantly, to explicitly compute its ice
nucleation rate by FFS method. At 250 K, the calculated
nucleation rate $9.6 \times 10^{30}$ m$^{-3}$ s$^{-1}$ for tetrahedral pyramid
exceeds those on flat graphene and in bulk water at the same
temperature (estimated on the basis of CNT[22]) by nearly 30 and
90 orders of magnitude, respectively.

The increasing degree of structural match through special
surface geometry also leads to an interesting enhancement of
polymorph selection for cubic ice $I_c$. Although scattered
experimental observations suggested the existence of cubic ice[23],
the unambiguous evidence of the well-defined cubic ice has not
been reported[24]. In fact, recent studies suggest that the commonly
referred cubic ice is indeed the stacking disordered ice[24–26]. It has
been shown that the ice freshly grown from either homogeneous
nucleation[21,27] or pre-existing hexagonal ice[28] is always stacking
disordered, with an overall cubicity within the range of 2/3 to 1/2,
depending on growth temperature. As shown in Fig. 2, when ice
crystallizes on flat graphene, it consists of randomly stacked ice
layers, with an overall fraction of cubic ice of 54% at 230 K. This is
also consistent with the cubicity obtained in homogeneous
nucleation[21]. When ice crystallizes within the 70° wedge, the
overall cubicity significantly increases to 82%, with only one
stacking fault separating two pieces of well-defined
$I_c$ crystals. Remarkably, ice is found to crystallize into nearly
pure $I_c$ within the tetrahedral pyramid, with an overall cubicity of
91%. The enhanced polymorph selection of ice is a natural
consequence of multi-dimensional structural match: a one-
dimensional match of ice basal plane applies no constraint to
the perpendicular stacking sequence, thus yielding a regular
stacking-disordered ice. In contrast, a two- or three-dimensional
match of intersecting ice basal plane aligns the ABCABC stacking
that is unique in $I_c$, because neither the ABAB stacking in $I_h$ nor
the disordered stacking allows two ice basal planes to intersect at
70° (or 110°). Figure 2 indeed shows that the increasing degree of
structural match not only enhances the overall cubicity, but also
significantly decreases the distribution variance, suggesting multi-
dimensional structural match can be an effective approach for
polymorph selection of cubic ice. Although beyond a certain size
cubic ice will turn into the normal hexagonal or stacking
disordered structure[24–26], the stability of the cubic structure can
now be preserved over a reasonably large size range, with the
assistance of this special geometry (see Supplementary Note 2 for
our test on a larger tetrahedral pyramid).

**Enhanced ice nucleation without matching ice lattice.** If the rate
enhancement at 70° and 110° can be well interpreted on the basis
of lattice match, then the sharp increase in the calculated ice
nucleation rate at $\beta = 45°$ appears rather intriguing, because
neither {0001} in $I_h$ nor {111} in $I_c$ forms a dihedral angle of 45°
with respect to another common lattice plane in either crystal. In
fact, examination of the crystallized ice structure (Fig. 1d) shows
that the first layers of ice on both wedge planes of the 45° wedge
are indeed ice basal planes, that is, same as in the 70° wedge.
Clearly, a mechanism other than lattice match is responsible for
rate enhancement.

Visualization of the crystallized ice within the 45° wedge
identifies a set of topological defects composed of coupled $5-7$
ring structure, as shown in Fig. 1e. The coupled $5-7$ ring
structure resembles the $5+7$ defect found in bulk ice[29]. As the
$5+7$ defect has a very long lifetime[30] and plays a substantial role

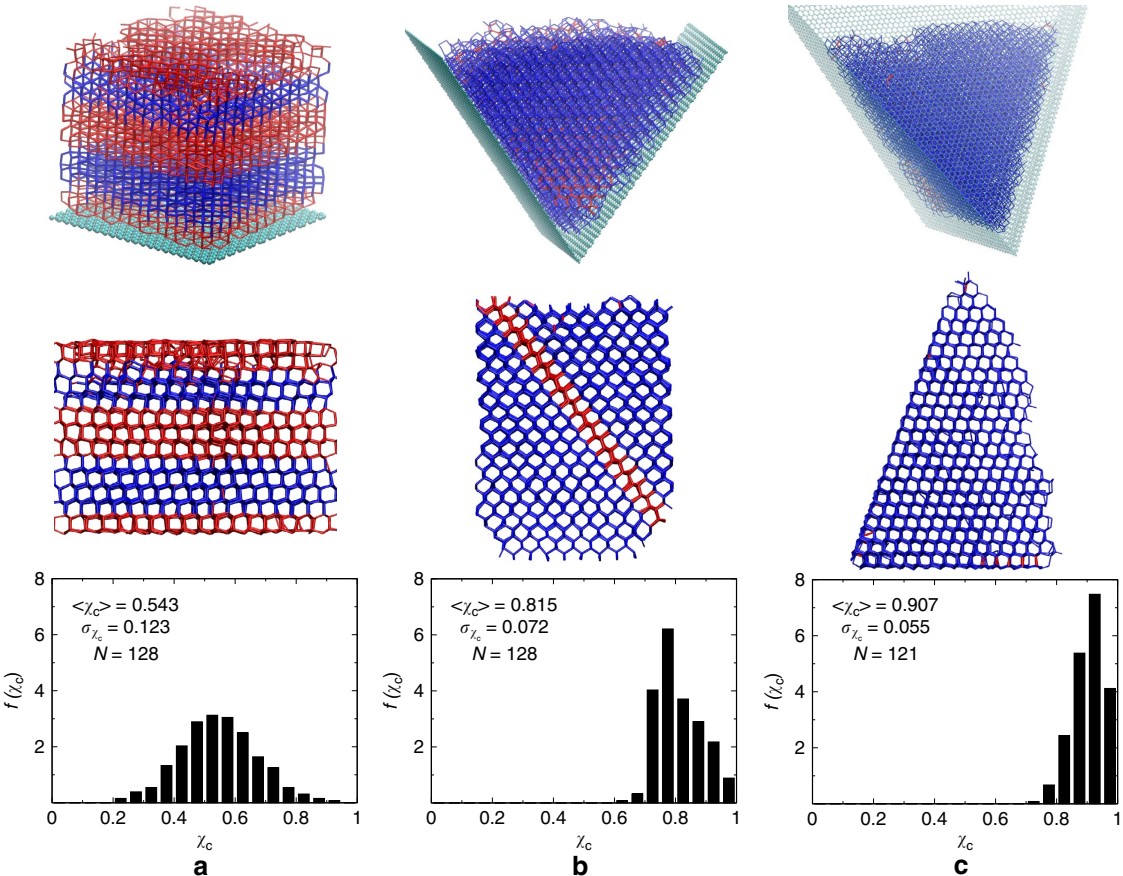

**Figure 2 | Enhanced polymorph selection of mW cubic ice $I_c$.** Ice exhibits a wide range of cubicity when it crystallizes (**a**) on a flat graphene, (**b**) within a 70° wedge and (**c**) within a tetrahedral pyramid. For each panel, the top and middle rows show the three-dimensional and side views of a representative configuration for fully crystallized ice, respectively. Red and blue represent the hexagonal $I_h$ and cubic $I_c$, respectively. The bottom row shows the calculated distribution of cubicity $\chi_c$, along with the corresponding mean cubicity $\langle\chi_c\rangle$, s.d. $\sigma_{\chi_c}$ and number of configurations $N$ used for computing the distribution. Cubicity $\chi_c$ is defined as the fraction of $I_c$ in ice $I$ (see Supplementary Note 1 for more details).

in mediating bulk melting of ice[30,31], it is not entirely surprising that the $5-7$ ring structure may also appear in the freezing of liquid water, because it can well be an intermediate state between liquid and solid. In fact, a similar structure does emerge in the direct molecular simulation of homogeneous ice nucleation based on the TIP4P water model[32]. It is then of interest to understand how this defect structure forms in the 45° wedge.

To answer this question, we examine the spontaneous crystallization trajectory within the 45° wedge, obtained at 230 K. As shown in Fig. 3a, the 45° wedge sees the frequent formation of a uniquely ordered structure at the wedge contact line, preceding the growth of regular ice lattice above it. This ordered structure can be regarded as the lateral repetition of a wedge-shape, non-ice-like building block, as shown in Fig. 3c. The wedge-shape core (WC) is composed of two six-membered rings that are connected by one hydrogen bond (HB) at the top and share two HBs at the bottom. The WC can be derived from the single diamond core (SDC), which is the building block of cubic ice $I_c$ (Fig. 3g), through replacing the two HBs on top of SDC by one HB. Therefore, the WC is akin to the crystalline phase of ice, but also involves a small distortion relative to the perfect tetrahedral network. In fact, the WC core has been previously identified as a stable fragment in the topological analysis of HB network in deeply supercooled water[33] and it is also found to form spontaneously in the 45° wedge when explicitly including HBs by employing an atomistic water model (see Supplementary Note 3).

Because of geometric constraint, the single WC core can only grow in parallel to the wedge contact line, by attaching other WC cores side-by-side. This parallel growth leads to a double WC (Fig. 3d), which provides the growth site for another polyhedral fragment on its top. As shown in Fig. 3e, this polyhedral cage $(6^5 5^2)$ is composed of five 6-membered rings as the prism planes and two 5-membered rings as the basal planes. In particular, the two 6-membered rings at the top of the cage are structurally compatible with the hexagonal core of $I_h$ and thus can serve as the anchoring point to grow regular hexagonal layers of ice parallel to graphene planes. As the growth of regular ice layers cannot geometrically fit the 45° wedge, pairs of $5-7$ rings emerge to bridge the structure gap. Interestingly, the $5-7$ topological defects are found to align themselves regularly, in analogy to a large-angle grain boundary (Fig. 1e).

Given the structural affinity between the WC and SDC, it is also of interest to compare the nucleation pathways of ice within the 70° wedge and the 45° wedge. As shown in Fig. 3g-j, the two nucleation pathways are indeed very similar at the early stage: both start with the formation of connected cores parallel to the wedge contact line. The difference is that in the 45° wedge, the filling of the second layer requires another topologically defective fragment, whereas in the 70° wedge, this can be achieved seamlessly by adding two intrinsic cubic ice building blocks on top. Interestingly, both pathways are found to be capable of strongly enhancing ice nucleation and they yield comparable nucleation rates at low temperature (Fig. 1a).

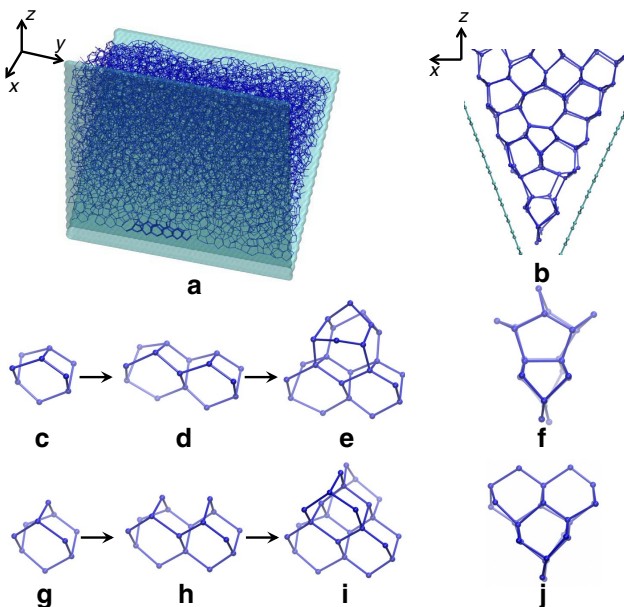

**Figure 3 | Molecular pathways of mW ice crystallizing near the tip of special wedges.** (**a**) Supercooled water (blue) at 230 K within the 45° wedge (green). The spontaneous formation of an ordered structure (highlighted) near the wedge contact line eventually leads to ice crystallization. Side view of ice is shown in **b**. (**c–e**) Nucleation pathway of ice within the 45° wedge. As a comparison, the nucleation pathway within the 70° wedge is shown in **g–i**. (**f,j**) Side views of the defect complex (**e**) and cubic ice complex (**i**), respectively.

Although an increasing temperature is found to gradually differentiate the two pathways, that is, the 45° wedge leads to a lower nucleation rate than the 70° wedge (Fig. 1b), both special geometries are still significantly more efficient than a flat surface for inducing ice formation (see Supplementary Fig. 4 and Supplementary Note 4).

## Discussion

The fact that both wedges lead to enhanced ice nucleation highlights the role of topological defects in ice nucleation. In fact, very little is known about the role defects play in nucleation[34]. Although topological defects were indeed found in homogeneous ice nucleation[21], current study explicitly shows that the formation of topological defects can directly promote ice nucleation. From a structural point of view, a topological defect can be viewed as an intermediate structure between liquid water and ice. It has been shown that water at low temperature is filled with stable topologically defective fragments with small distortions[33]. Therefore, in light of Ostwald step rule[35], it is not unexpected that the initial ice nucleation at moderate or high supercooling could also proceed with the formation of these defects as precursors. This may provide additional pathways to ice crystallization from a random HB network in liquid. However, as ice I consists of unique building blocks[36], in order for ice to grow, the topological defects must either transform themselves into ice cores directly (for example, by adding a HB to a WC, to become an SDC) or arrange themselves into a defect complex which is structurally compatible with the addition of regular ice cores. If an external medium could help facilitate the formation of such defect complexes through geometric constraint, just as in the 45° wedge, the nucleation of ice can be subsequently enhanced.

The enhanced ice nucleation within the 45° wedge can thus be rationalized by the following consideration: the geometrical constraint enforced by wedge tip significantly reduces the space that water molecules can explore and, if the motion of water molecules is restricted in a way compatible with a structural unit of ice, the crystallization of ice can be subsequently enhanced. This is equivalent to reducing the entropic part of the free energy barrier[17,19]. In this sense, all the enhancements of ice nucleation observed in this study, whether within the 45° wedge, 70°/110° wedge or tetrahedral pyramid, share the same thermodynamic rationale. The difference lies in the degree of such entropic barrier reduction, which depends on both the level of match (that is, zero-, one-, two- or three-dimensional) and the structure that is matched with.

Therefore, in essence, ice nucleation within the 45° wedge could be interpreted as a structural match at a two-dimensional level, albeit that instead of matching ice lattice directly, the wedge matches a topological defect complex, which in turn facilitates the growth of ice cores. Traditionally, the concept of structural match or templating effect in heterogeneous nucleation describes how well a nucleation centre is capable of inducing a structural ordering in liquid coherent with crystalline lattice. Our work suggests that this concept may be extended to include a broader structural match with non-crystalline units. An important implication for this extension is that when understanding or searching for a heterogeneous nucleation centre, the apparent lattice mismatch between a nucleation centre and the nucleus alone may not be used as a criterion to exclude a candidate from being an efficient nucleation centre. As there also exist a number of other metastable topological building blocks in liquid water, one would expect that some unconventional surface topography and structures could also lead to enhanced ice nucleation. Further studies are certainly needed to identify those structures and to understand their efficiency for inducing ice nucleation.

## Methods

**Molecular dynamics simulations.** Our MD simulations employes the mW[16] water model. The carbon–water interaction is represented by the two-body term of the mW model[10]. To create the configuration of water inside, we immerse the graphene wedge within a bulk water configuration and then remove water molecules outside the wedge. The configuration is relaxed to avoid any unphysical overlap of atoms and then equilibrated for 10 ns. Carbon atoms are frozen at the graphene lattice sites, thus their equations of motion are not integrated in our MD simulations. A periodic boundary condition is employed. The resulted simulation cell, as shown in Supplementary Fig. 1, typically involves about $4,500 \sim 7,500$ water molecules (with a volume of water of $134 \, nm^3 \sim 224 \, nm^3$), depending on the wedge angle $\beta$. The length of wedge contact line is 5.06 nm. The isothermal canonical ensemble (NVT) with a Nose-Hoover thermostat is employed throughout our simulations.

**Calculation of nucleation rate by forward flux sampling.** The nucleation rates of ice are computed by FFS[15] when ice nucleation becomes too slow to occur in direct MD simulations. The FFS has been successfully employed to study ice nucleation using both mW model[17,21,22,37,38] and atomistic water model[36,39]. In this approach, the nucleation trajectory is decomposed into a series of successive transitional segments on the basis of an order parameter $\lambda$. In the case of ice nucleation, such order parameter has been chosen and validated as the number of ice-like water molecules contained within the largest ice cluster[21,22,40]. The ice-like water molecule is numerically distinguished through the local order parameter $q_6$ such that a water molecule is considered to be truly ice-like when $q_6 > 0.5$ (ref. 21). In addition, the nearest neighbours of a truly ice-like water molecule is also considered as ice-like, to account for the interfacial layer separating ice and water. The rate constant $R$ is then obtained by the product of initial flux rate $\dot{\Phi}_{\lambda_0}$, which measures how frequently the system escapes from the basin $A$ (liquid) to reach the interface $\lambda_0$, and the growth probability $P(\lambda_B|\lambda_0)$ that evaluates how likely a configuration at interface $\lambda_0$ will eventually reach the basin $B$ (solid). Under the framework of FFS, the typically small $P(\lambda_B|\lambda_0)$ is calculated through $P(\lambda_B|\lambda_0) = \prod_{i=1}^{n} P(\lambda_i|\lambda_{i-1})$, where $P(\lambda_i|\lambda_{i-1})$ is the crossing probability for which a trajectory starts from the interface $\lambda_{i-1}$ and ends on the interface $\lambda_i$. $P(\lambda_i|\lambda_{i-1})$ can be directly obtained through firing a large number of trial runs at the interface $\lambda_{i-1}$. More details of application of FFS in ice nucleation can be found in refs 21,22.

**Estimating nucleation rate for spontaneous crystallization.** When crystallization becomes spontaneous in MD simulation, FFS is no longer needed to obtain crystallization trajectories. In such a case, nucleation rate can be obtained by firing multiple, independent MD shootings, through[41]:

$$R = \frac{N_C}{\left(\sum_{i=1}^{N_C} \tau_i + \sum_{j=1}^{N_{NC}} \tau_j\right) V}, \quad (1)$$

where $N_C$ is the number of crystallizing trajectories, $N_{NC}$ is the number of non-crystallizing trajectories, $\tau_i$ is the induction time for the $i$th crystallizing trajectory, $\tau_j$ is the trajectory length (simulation time) for the $j$th non-crystallizing trajectory and $V$ is the simulation volume. It is noted that heterogeneous nucleation rate should be measured by area (for nucleation on a surface) or length (for nucleation along a line). Here, a volume-based nucleation rate is used because it allows for a direct comparison between homogeneous and heterogeneous nucleation, and also because the volume of water is small and ice nucleation on wedge line or surface is strongly preferred.

Ice formation within the 43°, 45° and 47° wedges is found to be spontaneous within the order of $10^1$ ns from direct MD at 230 K. Therefore, multiple MD shootings are fired to calculate the corresponding ice nucleation rates using equation (1). To ensure convergence, the calculated rates are cross-checked against different numbers of crystallizing trajectories $N_C$ and the differences are found to be within 3% of the obtained rates. The details of these calculations are listed in Supplementary Table 1.

**Data availability.** The data that support the findings of this study are available from the corresponding author upon reasonable request.

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

## Acknowledgements

The work is supported by NSF through award CMMI-1537286. T.L. also thanks the Sloan Foundation through the Deep Carbon Observatory for supporting this work.

## Author contributions

T.L. conceived and designed the research project. Y.B. and T.L. performed and analysed the simulations. Y.B., B.C. and T.L. contributed to the interpretation of the results. Y.B. and T.L. wrote the paper.

## Additional information

**Competing interests:** The authors declare no competing financial interests.

