## [Peer Review File · Nature Communications]

Reviewers' comments:

Reviewer #1 (Remarks to the Author):

In their manuscript, Bi, Cao and Li report the results of rare event sampling methods of a minimal model of water to the study of heterogeneous ice nucleation on idealized surface geometries. The surfaces they study are formed from graphitic sheets in the shape of a wedge of varying angle. Using forward flux sampling and the mW model of water, they find that the ice nucleation rate can vary nonmonotonically with the wedge angle, a result that is in opposition to simple extensions to classical nucleation theory. The manuscript is well written and easy to understand, the results are interesting, and the numerical work convincing enough to warrant publication in Nature Communications. At present, however, the manuscript suffers from incomplete and unsatisfying discussion the authors observations. I think that the work would benefit greatly from some additional analysis, that aims to explain in more quantifiable terms, why the nucleation rate changes with angle in the way that it does.

A few outstanding questions along this direction are listed below.

1) Can the authors quantify the role of the topological defects responsible for rate enhancement for wedges near 45 o? Currently I find the authors argument for the importance of these defects unsatisfying. They are observed en route to ice formation, but do they actually decrease the relevant free energy barrier? Alternatively, can the authors quantify their mechanistic importance by evaluating commitor probabilities?

2) What is the role of strain in nucleating ice in wedges that do not exactly match the ice lattice? Does this energetic penalty account for the decrease around 70 and 110 degrees? If not can the authors formulate an expression or explanation for the form of the rate enhancement around these observed maximums based on something else (like the ratio of nucleation events at the apex of the wedge compared to the sides)?

3) Is there a simple explanation for the relative rate enhancement and specifically its temperature dependence?

Reviewer #2 (Remarks to the Author):

Bi, Cao and Li study the freezing of a simple (mW) model of water, in graphene/graphite wedges. This is done using computer simulations, and the FFS technique. They find that as the wedge angle is decreased the nucleation rate tends to increase, as the standard theory (classical nucleation theory) predicts. But they also find that the variation is not monotonic, there are maxima in the rates. A similar maximum has been found for a model of argon, but this finding is new for ice. They also find that wedges alters the ice polymorph that forms. This sort of polymorph control has not been shown for any system before, as far as I am aware. It is also very interesting.

I also enjoyed their discussion of how a specific defect forms in wedges with 45 degree wedge angles.

The paper clearly shows novel and potentially important behaviour in a system of great interest - the authors' President-elect is wrong to apparently think that global warming is a not real. It is real and cloud models with accurate ice nucleation predictions are needed to model it. Concave carbon surfaces could be significant in ice nucleation in clouds under some circumstances.

So, I am happy to recommend publication. Just a few suggestions:

1) Presentation. The manuscript is generally clear and well written, I assume that English is not

the first language of any of the authors, but this rarely shows, I am impressed. However it has no sections. I would say that it is too long not to have sections, adding sections such as: Introduction, Model and Simulation, Results for nucleation in wedge, etc, would significantly help the reader. I believe headings and sub-headings are allowed by Nature Comms.

2) The authors' idea for why the rate has maxima near 70 and 110 degrees, and why wedges favour the cubic polymorph (that at these angles a piece of cubic ice bounded by two 111 planes which have low surface free energies when in contact with the wedge walls, fits perfectly into the wedge) is good. But does the absence of the nucleation of hexagonal ice imply that these same planes in hexagonal ice can only exist parallel to each other? I.e., that are there no angles < 180 degrees where a piece of hexagonal ice will fit into a wedge, with these same planes in contact with the walls?

If this true, then the dominance of cubic ice found by the authors would be expected, if it is not true then it is puzzling that no hexagonal ice appears at whatever is the angle < 180 degrees that the hexagonal ice nucleus will fit in the wedge.

Could the authors comment on this?

3) Ice nucleation, especially at the low temperatures (230K) considered by the authors, can be difficult to simulate due to strong hydrogen bonds introducing slow microscopic kinetics. Now the mW model the authors use, makes life easier in this respect, but it would reassuring to a test that FFS is sampling correctly.

Possible tests include:

i) Changing how lambda is defined a little bit, perhaps to say deliberately favour hexagonal or cubic ice, to check for order parameter sensitivity.

ii) Doing direct simulations, at temperatures a little lower so the nucleation barrier is lower and FFS is not needed. For example, to see if the bias to cubic ice, and the 5-7 defects, that the authors have seen with FFS, survive. I do not know how practicable this with mW, presumably at very low temperatures the system will arrest on simulation timescales, but at least at the angles where the rate is maximal, this may work in the sense some direct simulations are possible. These could be done in a smaller system.

iii) Umbrella sampling simulations, or some other method for rare events.

There are other possible checks. But evidence or an argument that FFS sampling is not biasing the crystals that form, would be good. Note one method would be enough, I am not asking for i), ii) and iii), just one of them or an alternative!

4) Their FFS simulations must have many, presumably 100s or more, of configurations, as this number is needed to sample enough to get reliable rates. With this in mind the authors have much more data than is given in, say Figure 2. For example, in Figure (b) they quote a figure of 82% cubicity, and there is one clear stacking fault in the snapshot. As they have a 100+ final configurations, it would be good to know the distributions of cubicity amongst these, for example how wide is the distribution? 70 to 90%, 40 to 95%,...?

Is this distribution particularly narrow and with particularly narrow distributions at the angles where the rate is maximal? I.e., how does it vary with wedge angle, and between the wedge and the tetrahedral-pyramid pore?

5) Although I am not an expert on the ice lattices, as far as I can tell their analysis of how nucleation occurs in wedges with 45 degree angles looks convincing. I find their discussion of how

the ice forms very interesting, and I agree with the authors' conclusion that this formation via defects-optimised-for-that-angle is something we should consider in future work.

One comment: earlier work on a model for argon (their ref [11]) found that at the smallest angles the rate decreases, which is what they find, simply because in the tip of very narrow wedges the molecules could not crystallise, resulting in an always-liquid tip which inhibited crystallisation. Is an always-liquid region near the tip of the wedge also found in their systems with wedge angles narrower than 30 to 40 degrees - where the nucleation rate falls dramatically?

Reviewer #3 (Remarks to the Author):

This is an interesting paper that demonstrates that simple surfaces of different shape can influence ice nucleation. This work builds on previous work with other graphite surfaces, and on earlier work which showed similar effects for simple atomistic models. Nevertheless, it is useful to show that similar effects occur for waterlike models, as is done in this paper. Generally, the paper is clearly written and the arguments given are substantiated by the simulation analysis. In some places the inclusion of articles such as "the", "a", or "an" would make the paper a little easier to read. Overall, in my opinion this paper merits publication, but I do have one important question that should be addressed.

Both in the paper (page 12) and in the Supp. information the authors mention that simulations were carried out for both the atomistic Mw water model, and TIP4P/Ice, but as far as I can see no direct comparisons are made. Also, it is not totally clear to me if the results plotted were obtained with the Mw model or with TIP4P/Ice. This should be clarified. Also, I would like to see some direct comparisons between the two models. The Mw model is fundamentally atomistic with the water geometry imposed by three-body interactions, whereas TIP4P/Ice is much more realistic in that electrostatic forces are included. It is important to demonstrate that both models show a similar sensitivity to the wedge angle etc. in order to judge how well the simulation results might apply to real water.

We thank all the referees, for their interest in our work, for their careful reading, and for their constructive and interesting comments that prompted us to carry out additional studies and analyses. Below we address all the comments and questions raised by the reviewers in the order as appeared in the report.

Response to Reviewer #1's comments:

- 1) “Can the authors quantify the role of the topological defects responsible for rate enhancement for wedges near 45 o? Currently I find the authors argument for the importance of these defects unsatisfying. They are observed en route to ice formation, but do they actually decrease the relevant free energy barrier? Alternatively, can the authors quantify their mechanistic importance by evaluating commitor probabilities?”

We thank Reviewer #1 for this comment. In fact we had indeed considered quantifying the role of topological defects observed in the 45-degree wedge, particularly on its effect on changing nucleation barrier. However we found there exist a few fundamental issues refraining us from conducting an exact study of this nature. First, when referring to a decrease of free energy barrier, one needs to first define a meaningful reference, i.e., to understand the free energy barrier is reduced relative to which reference pathway. For the purpose of understanding the role of topological defects, a rational choice of such reference pathway would be an alternative pathway that *does not involve topological defects but coexists under the same temperature and same degree of geometrical constraints*. Unfortunately, such pathway does not exist near the contact line of the 45-degree wedge, because the geometric constraint enforced by the 45-degree wedge does not allow the topologically perfect ice structure to form near the wedge contact line. It should be noted that topologically perfect ice could certainly form away from the contact line, e.g., on the planar part of wedge plane, or even within the bulk region of water. However, in those cases it is no longer a fair comparison, because water molecules are under different degrees of geometrical constraints (i.e., 0D for homogeneous/1D for flat surface vs. 2D for wedge). In fact, in those cases the nucleation of ice already becomes many orders of magnitude **slower**, and it requires FFS to obtain the rates and trajectories. In contrast, ice crystallization within the 45-degree wedge was found to occur **spontaneously and frequently in brute-force MD**, and the nucleation is indeed found to begin by the formation of topological defects near the contact line.

A possibly meaningful reference, although not perfect, could be the ice nucleation pathway within the 70-degree wedge, because in both cases, ice crystallizes in comparable environments and their molecular pathways are very similar. To estimate the difference in their free energy barriers, one could calculate the free energy profile as a function of an order parameter. Although expensive, we have indeed managed to obtain these profiles recently for homogeneous ice nucleation and hydrate nucleation (see JCP 145, 211909 (2016)) on the basis of FFS. However we are uncertain whether we could obtain a reliable estimate of such difference in the current study, because the difference in the rates between the 45-degree and 70-degree wedge is very small (about a factor of three) at 230 K (see Fig. 1 of the revised manuscript, and Supplementary Fig. S4). This means the difference in the barrier can be smaller than $1 kT$, which is well within the statistical uncertainty of nearly any approach for estimating free energy profile.

In spite of the closeness in the free energy barriers, we feel that the rate of nucleation is probably a better, and perhaps a more direct, quantitative measure for the relevance of a nucleation pathway. After all, the *directly calculated* rate, either through direct MD or FFS, is independent of any nucleation theory, and thus the key metric to understand the relevance of a nucleation mechanism. Therefore, although we have not been able to obtain a quantitative description in terms of the change of nucleation barrier when topological defects form, we do feel that the obtained evidences, including

both extremely high nucleation rate and the fact that the spontaneous nucleation is directly induced by defects, strongly indicate the active role of these defects.

On a related note, we can also rationalize the enhanced ice nucleation within the 45-degree wedge by the following consideration: The geometrical constraint enforced by the 45-degree wedge significantly reduces the space that water molecules can explore, because the motion of water molecules is restricted near the tip, and the restriction is compatible with a metastable structural unit of ice. This is equivalent to reducing the entropic barrier of nucleation. In this sense, all the enhanced nucleation observed in this study, whether within the 70-degree/110-degree, 45-degree, or pyramid, has the same thermodynamic rationale. The difference lies in the degree of reduction in such entropic barrier, which depends on both the level of match (i.e., 2-D or 3-D) and the structure that is matched with (i.e., crystalline or non-crystalline). We believe this is the direction of our future study for further rationalizing heterogeneous nucleation. We have also included this discussion in the revised manuscript in the Discussion section (page 12).

2) *“What is the role of strain in nucleating ice in wedges that do not exactly match the ice lattice? Does this energetic penalty account for the decrease around 70 and 110 degrees? If not can the authors formulate an expression or explanation for the form of the rate enhancement around these observed maximums based on something else (like the ratio of nucleation events at the apex of the wedge compared to the sides)?”*

We fully agree with Reviewer #1 that the decrease of ice nucleation rate around 70 and 110 degrees is due to strain effect, which introduces additional strain-energy penalty that nucleation would have to overcome when wedges do not match exactly ice lattice. This is equivalent to increasing the enthalpic part of nucleation barrier. In fact in our previous study (JPCC 120, 1507 (2016)) we have explicitly investigated the role of strain (lattice mismatch) in heterogeneous ice nucleation rate. We thus include a brief explanation in the revised manuscript (on page 4).

3) *“Is there a simple explanation for the relative rate enhancement and specifically its temperature dependence?”*

We want to thank Reviewer #1 for pointing out the possible temperature dependence. Motivated by this comment, we performed additional rate calculations for both 45-degree and 70-degree wedges, as a function of temperature between 230 K and 240 K. Interestingly but not surprisingly, the calculation indicates a crossover in the rate near 236 K, above which the 70-degree wedge exhibits a stronger ice nucleation efficiency than the 45-degree wedge. Although a future study is needed to further understand this trend, we believe this temperature dependence is originated from the high stability and propensity of topological defects. As the previous study (JCP 127, 134504 (2007), Ref. 34) suggested, the topological fragments become increasingly stable as temperature decreases, and nearly all the hydrogen-bond network is covered by these fragments at low temperature. The high stability and abundance of these topological defects make them kinetically more accessible to crystallization pathways at low temperature, which is in consistent with the Ostwald step rule. On the contrary, the increasing temperature leads to a decrease in both stability and propensity of topological defects, thus making this alternative nucleation route less favorable than a direct pathway in the 70-degree wedge. However in this case it should be noted that the pathway in the 45-degree wedge is still strongly more favorable than that on a flat surface, because ice nucleation within the 45-degree wedge is still facilitated by a 2D-level match rather than by a 1D match on flat surface.

We have included this new study and analysis in the revised manuscript (page 11) and Supplementary Information (Note 5 and Fig. S4).

Response to Reviewer #2's comments:

- 1) *“Presentation. The manuscript is generally clear and well written, I assume that English is not the first language of any of the authors, but this rarely shows, I am impressed. However it has no sections. I would say that it is too long not to have sections, adding sections such as: Introduction, Model and Simulation, Results for nucleation in wedge, etc, would significantly help the reader. I believe headings and sub-headings are allowed by Nature Comms.”*

We are grateful for this nice comment by Reviewer #2's. We have also followed Reviewer #2's suggestions by adding the corresponding sections to the revised manuscript.

- 2) *“The authors' idea for why the rate has maxima near 70 and 110 degrees, and why wedges favour the cubic polymorph (that at these angles a piece of cubic ice bounded by two 111 planes which have low surface free energies when in contact with the wedge walls, fits perfectly into the wedge) is good. But does the absence of the nucleation of hexagonal ice imply that these same planes in hexagonal ice can only exist parallel to each other? I.e., that are there no angles < 180 degrees where a piece of hexagonal ice will fit into a wedge, with these same planes in contact with the walls? If this true, then the dominance of cubic ice found by the authors would be expected, if it is not true then it is puzzling that no hexagonal ice appears at whatever is the angle < 180 degrees that the hexagonal ice nucleus will fit in the wedge. Could the authors comment on this?”*

We thank Reviewer #2 for finding our interpretation of structural match reasonable. The basal planes of hexagonal ice {0001} indeed do not intersect with each other. It is also true that the wedge, *if its*

Figure 1: Fully crystallized ice within the 135-degree wedge. Color codes: red (Ih), blue (Ic), yellow (liquid-like water), green (carbon).

both surfaces only grow ice basal plane simultaneously, cannot geometrically fit a **single crystal** hexagonal ice. However this does not necessarily mean hexagonal ice cannot form within the wedge, because hexagonal structure can always grow subsequently on top of cubic ice, with the assistance of stacking fault. In fact the obtained ice structure for those non-structural matching wedges, e.g., the 135-degree wedge (see Figure 1), shows the existence of hexagonal ice, albeit in the form of stacking disorder. This is also what typically occurs in homogeneous nucleation and heterogeneous nucleation on flat graphene surface.

On a side note: A possibility to grow a single crystal hexagonal ice through an atomically sharp wedge is to form a wedge by *two surfaces that both promote the growth of prism faces {10 $\bar{1}$ 0} of ice*. Since the dihedral angle between two prism faces is 60°/120°, a 60-degree wedge geometrically fits the hexagonal ice perfectly. Just as a 70°/110° basal-plane-promoting wedge promotes cubic ice, it is expected that a 60°/120° prism-face-promoting wedge can significantly promote both ice nucleation rate and polymorph selection for pure hexagonal ice Ih. This may be possibly tested in future simulation work. We also believe the multi-dimensional geometric match approach, if realized in experiment, could provide a new strategy for controlling nucleation and polymorph selection, and its applicability is certainly not limited to ice.

- 3) *“Ice nucleation, especially at the low temperatures (230K) considered by the authors, can be difficult to simulate due to strong hydrogen bonds introducing slow microscopic kinetics. Now the mW model the authors use, makes life easier in this respect, but it would reassuring to a test that FFS is sampling correctly. Possible tests include: i) Changing how lambda is defined a little bit,*

perhaps to say deliberately favour hexagonal or cubic ice, to check for order parameter sensitivity. ii) Doing direct simulations, at temperatures a little lower so the nucleation barrier is lower and FFS is not needed. For example, to see if the bias to cubic ice, and the 5-7 defects, that the authors have seen with FFS, survive. I do not know how practicable this with mW, presumably at very low temperatures the system will arrest on simulation timescales, but at least at the angles where the rate is maximal, this may work in the sense some direct simulations are possible. These could be done in a smaller system. iii) Umbrella sampling simulations, or some other method for rare events. There are other possible checks. But evidence or an argument that FFS sampling is not biasing the crystals that form, would be good. Note one method would be enough, I am not asking for i), ii) and iii), just one of them or an alternative!"

We absolutely agree with Reviewer #2 that although FFS is a powerful approach, extra care should be taken to ensure its sampling correctness. In fact *ice crystallization around the 45-degree wedge at 230 K actually occurs spontaneously in direct MD simulation (within a time scale of 10^1 ns)* when the initial configurations were prepared for FFS sampling. Therefore the nucleation rate was high enough that FFS was in fact **not** needed for modeling ice crystallization for these wedges at 230 K. Correspondingly, the nucleation rates at 230 K for wedges around 45 degree *in the original manuscript (Fig. 1a in the original manuscript)* were estimated based on the induction time of a few crystallization trajectories. While brief, this should give a good estimate of the order of magnitude of ice nucleation rate.

To improve the quality of these estimates, particularly in order to obtain a more reliable, *quantitative* measure for these rates, we carried out a large number of direct MD shootings for the 43-degree, 45-degree, and 47-degree wedges at 230 K. The rates can be determined by the number of crystallization events, the total induction time, and the total length of non-crystallizing trajectories. We have also carried out additional analysis to ensure these directly obtained rates are well converged. These new results, along with the details of simulation and analysis, are reported in the revised Supplementary Information (Supplementary Note 1 and Table S1). The corresponding text (page 5, 13) and Fig. 1a in the main text was also modified to reflect this update. These unbiased direct simulation results re-ensure our main conclusions are robust.

4) *"Their FFS simulations must have many, presumably 100s or more, of configurations, as this number is needed to sample enough to get reliable rates. With this in mind the authors have much more data than is given in, say Figure 2. For example, in Figure (b) they quote a figure of 82% cubicity, and there is one clear stacking fault in the snapshot. As they have a 100+ final configurations, it would be good to know the distributions of cubicity amongst these, for example how wide is the distribution? 70 to 90%, 40 to 95%,...? Is this distribution particularly narrow and with particularly narrow distributions at the angles where the rate is maximal? I.e., how does it vary with wedge angle, and between the wedge and the tetrahedral-pyramid pore?"*

We clarify that our FFS simulations did generate over 100 configurations at each interface, but in practice, our FFS simulation is usually stopped after the barrier has been well crossed (for example, 200~300 molecules larger than the critical size), i.e., when the crossing probability becomes one so that crystallization turns into a spontaneous process already. Therefore our original FFS simulations had not directly yielded large statistics for *fully crystallized configurations* yet.

Nevertheless, to address the interesting question raised by Reviewer #2, we extended our FFS simulations on flat surface (230 K), 70 degree wedge (230 K), and tetrahedral pyramid (250 K), to completely crystallize the simulation cell, by firing shootings near the critical size so that about half of the shootings resulted in full crystallization. We then analyze the distribution of cubicity based on the obtained ensemble of configurations. Indeed, we found the increasing degree of structural match *not*

only enhances the overall cubicity, but also significantly decreases the distribution variance. This suggests that enhancing the degree of structural match can be an effective approach for controlling polymorph selection. We have included these new results and discussion in the revised manuscript on page 7 and Fig. 2, and Supplementary Information Note 2.

5) *“Although I am not an expert on the ice lattices, as far as I can tell their analysis of how nucleation occurs in wedges with 45 degree angles looks convincing. I find their discussion of how the ice forms very interesting, and I agree with the authors' conclusion that this formation via defects-optimised-for-that-angle is something we should consider in future work. One comment: earlier work on a model for argon (their ref [11]) found that at the smallest angles the rate decreases, which is what they find, simply because in the tip of very narrow wedges the molecules could not crystallise, resulting in an always-liquid tip which inhibited crystallisation. Is an always-liquid region near the tip of the wedge also found in their systems with wedge angles narrower than 30 to 40 degrees - where the nucleation rate falls dramatically?”*

We thank Reviewer #2 again for the praise. Regarding the structure of ice near the tip, what we found is similar to the early study (Ref. 14) when wedge does not fill ice lattice geometrically. In this scenario, ice does not nucleate near wedge tip, but rather, on one of the graphene planes far away from the wedge tip. Not surprisingly, the corresponding nucleation rate becomes essentially same as that on a flat surface. As ice crystallizes, it leaves a non-ice like (or liquid like) structure near the tip and near the other graphene plane, just as shown in Figure 1.

Response to Reviewer #3's comments:

1) *“This is an interesting paper that demonstrates that simple surfaces of different shape can influence ice nucleation. This work builds on previous work with other graphite surfaces, and on earlier work which showed similar effects for simple atomistic models. Nevertheless, it is useful to show that similar effects occur for waterlike models, as is done in this paper. Generally, the paper is clearly written and the arguments given are substantiated by the simulation analysis. In some places the inclusion of articles such as “the”, “a”, or “an” would make the paper a little easier to read. Overall, in my opinion this paper merits publication, but I do have one important question that should be addressed. Both in the paper (page 12) and in the Supp. information the authors mention that simulations were carried out for both the atomistic Mw water model, and TIP4P/Ice, but as far as I can see no direct comparisons are made. Also, it is not totally clear to me if the results plotted were obtained with the Mw model or with TIP4P/Ice. This should be clarified. Also, I would like to see some direct comparisons between the two models. The Mw model is fundamentally atomistic with the water geometry imposed by three-body interactions, whereas TIP4P/Ice is much more realistic in that electrostatic forces are included. It is important to demonstrate that both models show a similar sensitivity to the wedge angle etc. in order to judge how well the simulation results might apply to real water.”*

We thank Reviewer #3 for finding our work interesting. We have followed the suggestion by carefully proofreading our manuscript, with a special focus on the usage of articles. To avoid any possible confusion regarding water models, we have also explicitly clarified “mW water” or “mW ice” throughout the revised manuscript. Since the main text of our manuscript only discusses the results based on the mW water, we have also removed the phrase “TIP4P/Ice model” in the Methods section (page 13).

We also agree with Reviewer #3 that the obtained findings/results also need to be understood in the context of the water models used. This is precisely why we carried out a study on the 45-degree wedge by employing the atomistic water model TIP4P/Ice, and choosing the same temperature and water-

surface interaction strength as used in the mW case. Although no spontaneous ice crystallization was found within a run of 50 ns, we indeed have observed the frequent formation of the key structural motif when atomistic water model is employed, as reported in Supplementary Note Four.

With regard to a *direct* comparison between the two models, as much as we would love to have such comparison, our past experience has taught us that this is a very challenging task. In fact the PI of this study has started the effort to freeze atomistic water based on FFS long time ago—right after he successfully did so for Si and Ge in 2008. During the course of this effort, we learned that although FFS is a powerful sampling approach, its success relied on massive shooting trials which are inevitably computationally expensive. When FFS is applied to an atomistic water model, the slow dynamics of explicit hydrogen-bonded network makes it extremely demanding to freeze atomistic water. This is not surprising, as according to a benchmark made by Valeria Molinero, mW model is about 180 times faster than the SPC/E model (JPCB 113, 4011, 2009). In fact the first direct rate calculation of homogeneous ice nucleation based on an atomistic water model (TIP4P/Ice) by FFS had not been achieved until 2015 (by Pablo Debenedetti's group, PNAS 112, 10582), and this did not come easily—it cost over *21 million CPU hours*. This translates into a 24/7, five-year computation on our group-own HPC with 432 CPU cores. Certainly, a higher rate of heterogeneous nucleation may alleviate the problem if the condition and surface are carefully chosen, but its cost can still be prohibitively high. In fact in the recent study (where the PI of this manuscript was involved) to model heterogeneous ice nucleation on Kaolinite surface (JPCL 7, 2350 (2016), the calculation still cost about 6.7 million CPU hours.

Fortunately, the monoatomic model of water mW has been shown to provide a good description of water/ice system, not only on its thermodynamics, but also on structures, as revealed by many recent studies. In particular, the predicted cubicity and stacking disorder of mW is found to be in an excellent agreement with experiments (e.g., PCCP 17, 60 (2015)). In addition, the spontaneous occurrence of the key structural motifs when both mW and TIP4P/Ice were employed, along with the fact that the metastable topological defects have been previously identified in the *atomistic water model* (six-site model), gives us confidence that the obtained results should apply to real water.

REVIEWERS' COMMENTS:

Reviewer #1 (Remarks to the Author):

In their response, the authors have adequately addressed my concerns. The manuscript as it is currently written is suitable for publication in Nature communications.

Reviewer #2 (Remarks to the Author):

I would like to thank the authors for their work in responding to my comments. They have improved an already good paper, and answered my questions. I am very happy to recommend publication.

I have only one further comment, and I am sorry I did not spot this in the first round. The authors consistently express rates as per unit volume (and per unit time). Now, rates of homogeneous nucleation are naturally expressed per unit volume, as the total rate of homogeneous nucleation scales with volume, for large volumes. However, if nucleation is dominated by nucleation along the line of a wedge, then for the deep wedges, the nucleation rate scales with wedge length not volume.

Consistently using rates per unit volume allows comparison between homogeneous and heterogeneous nucleation rates, but much of the earlier work on nucleation in wedges used rates per unit length. In addition the rate per unit volume of a wedge depends on wedge volume, and they don't seem to specify the volume, it is not in the methods section which just describes calculation of a rate (i.e. quantity with units of one over time).

Could the authors specify the volume they use? And ideally the wedge length so the rate per unit length could be determined by interested readers.

Reviewer #3 (Remarks to the Author):

The authors have addressed the questions that I raised in my earlier detailed review. I recommend that the paper now be published.

We thank all three reviewers for finding our revision satisfactory.

Response to Reviewer #2's comments:

“I would like to thank the authors for their work in responding to my comments. They have improved an already good paper, and answered my questions. I am very happy to recommend publication. I have only one further comment, and I am sorry I did not spot this in the first round. The authors consistently express rates as per unit volume (and per unit time). Now, rates of homogeneous nucleation are naturally expressed per unit volume, as the total rate of homogeneous nucleation scales with volume, for large volumes. However, if nucleation is dominated by nucleation along the line of a wedge, then for the deep wedges, the nucleation rate scales with wedge length not volume. Consistently using rates per unit volume allows comparison between homogeneous and heterogeneous nucleation rates, but much of the earlier work on nucleation in wedges used rates per unit length. In addition the rate per unit volume of a wedge depends on wedge volume, and they don't seem to specify the volume, it is not in the methods section which just describes calculation of a rate (i.e. quantity with units of one over time). Could the authors specify the volume they use? And ideally the wedge length so the rate per unit length could be determined by interested readers.”

We absolutely agree with Reviewer #2 that heterogeneous nucleation rate should scale with area (on a surface) or length (one a line), not with volume. We chose the volume-based rate instead of area-based or length-based rate in this study (also in our previous study, e.g., PRE 91, 052402 (2015)), because it conveniently allows comparing directly between homogeneous and heterogeneous nucleation, just as mentioned by Reviewer #2. For clarification, in Methods section we provide a short explanation of this choice, along with both the volume (134~224 nm³) and wedge length (5.06 nm) used in our simulations. Those interested readers can easily determine the corresponding rate per unit length.